# Cadmium-Inspired Self-Polymerization of {Ln^III^Cd_2_} Units: Structure, Magnetic and Photoluminescent Properties of Novel Trimethylacetate 1D-Polymers (Ln = Sm, Eu, Tb, Dy, Ho, Er, Yb)

**DOI:** 10.3390/molecules26144296

**Published:** 2021-07-15

**Authors:** Maxim A. Shmelev, Ruslan A. Polunin, Natalia V. Gogoleva, Igor S. Evstifeev, Pavel N. Vasilyev, Alexey A. Dmitriev, Evgenia A. Varaksina, Nikolay N. Efimov, Ilya V. Taydakov, Aleksey A. Sidorov, Mikhail A. Kiskin, Nina P. Gritsan, Sergey V. Kolotilov, Igor L. Eremenko

**Affiliations:** 1N. S. Kurnakov Institute of General and Inorganic Chemistry, Russian Academy of Sciences, 31 Leninsky Prosp., 119991 Moscow, Russia; shmelevma@yandex.ru (M.A.S.); judiz@rambler.ru (N.V.G.); i.evstifeev@gmail.com (I.S.E.); anubisvas@gmail.com (P.N.V.); nnefimov@yandex.ru (N.N.E.); sidorov@igic.ras.ru (A.A.S.); ilerem@igic.ras.ru (I.L.E.); 2L. V. Pisarzhevskii Institute of Physical Chemistry, National Academy of Sciences of Ukraine, 31 Prosp. Nauki, 03028 Kiev, Ukraine; ingoldp@mail.ru (R.A.P.); s.v.kolotilov@gmail.com (S.V.K.); 3V. V. Voevodsky Institute of Chemical Kinetics and Combustion, 3 Institutskaya Str., 630090 Novosibirsk, Russia; dmitralexey@gmail.com (A.A.D.); nina.gritsan@gmail.com (N.P.G.); 4Novosibirsk State University, 2 Pirogova Str., 630090 Novosibirsk, Russia; 5P. N. Lebedev Physical Institute, Russian Academy of Sciences, 53 Leninsky Prosp., 119991 Moscow, Russia; janiy92@yandex.ru (E.A.V.); taidakov@mail.ru (I.V.T.); 6Academic Department of Innovational Materials and Technologies Chemistry, Plekhanov Russian University of Economics, 117997 Moscow, Russia

**Keywords:** cadmium(II), lanthanide(III), heterometallic complexes, coordination polymers, pivalic acid, X-ray diffraction study, magnetochemistry, photoluminescence, *ab initio* calculations

## Abstract

A series of heterometallic carboxylate 1D polymers of the general formula [Ln^III^Cd_2_(piv)_7_(H_2_O)_2_]_n_·nMeCN (Ln^III^ = Sm (**1**), Eu (**2**), Tb (**3**), Dy (**4**), Ho (**5**), Er (**6**), Yb (**7**); piv = anion of trimethylacetic acid) was synthesized and structurally characterized. The use of Cd^II^ instead of Zn^II^ under similar synthetic conditions resulted in the formation of 1D polymers, in contrast to molecular trinuclear complexes with Ln^III^Zn_2_ cores. All complexes **1**–**7** are isostructural. The luminescent emission and excitation spectra for **2**–**4** have been studied, the luminescence decay kinetics for **2** and **3** was measured. Magnetic properties of the complexes **3**–**5** and **7** have been studied; **4** and **7** exhibited the properties of field-induced single-molecule magnets in an applied external magnetic field. Magnetic properties of **4** and **7** were modelled using results of SA-CASSCF/SO-RASSI calculations and SINGLE_ANISO procedure. Based on the analysis of the magnetization relaxation and the results of *ab initio* calculations, it was found that relaxation in **4** predominantly occurred by the sum of the Raman and QTM mechanisms, and by the sum of the direct and Raman mechanisms in the case of **7**.

## 1. Introduction

The design and synthesis of new coordination compounds with two or more different physical properties that are promising for practical application, as well as the search for the ways to modify the physical properties (including their reversible change) are urgent problems of modern coordination chemistry and physical chemistry [1,2,3]. Such properties may include a combination of non-trivial magnetism and luminescence [4,5] or conductivity [6,7,8], a combination of magnetism and rotation of polarized light [9,10], sensitivity of magnetic properties to irradiation [11], or thermochromic properties [12], electrochromism [13], the ability to convert mechanic deformation into voltage and vice versa [14] to display thermoelectric behavior [15] or magnetocaloric effect [16]. A change in the physical properties of a compound in comparison with the known prototype can be achieved by fine-tuning its structure by changing the synthesis conditions [17] or by an irreversible post-synthetic reaction [18,19], while a reversible change in the property can be caused by interaction with a certain substrate [20].

The choice of Ln^III^ compounds for this study was caused by the unique properties of such ions, i.e., their ability to exhibit slow magnetic relaxation (the so-called Single-Molecule Magnetism, SMM, or Single Ion Magnetism, SIM) [21] along with luminescence [22]. A lot of coordination polymers, containing only Ln^III^ ions as paramagnetic centers, exhibit SMM properties (in particular, slow relaxation of magnetization) [23,24,25], however, in many cases, the role of the exchange coupling between Ln^III^ ions (if any) in the occurrence of SMM properties was not clear. In some cases, it was shown that exchange interactions between paramagnetic ions can facilitate magnetic relaxation via the mechanism of quantum tunneling of magnetization (QTM) [26], but the opposite effect was also found: QTM was quenched due to the Cr^III^···Dy^III^ coupling compared to the analog containing diamagnetic Co^III^ [27]. More interesting effects were found for magnetically diluted Ln complexes, characterized by slow magnetic relaxation [23,28]. In addition, long-range magnetic ordering was found in the Ln^III^ coordination polymers [29]. 

Thus, the main goal of this study was to reveal the effect of diamagnetic dilution of Ln^III^ ions with Cd^II^ on the magnetic properties. Dilution of paramagnetic complexes inside an isostructural diamagnetic matrix reduces dipolar interactions and suppresses QTM, allowing examining the individual behavior of a metal ion in a particular environment [30,31,32]. Notably, it was shown that diamagnetic ions can induce a change in the magnetic properties of Ln^III^ due to a change in the distortion of its coordination polyhedron [4,33]. A direct effect of Cd^II^ on the Ln^III^ luminescence is not expected; there was no reason to expect the suppression of luminescence due to the incorporation of Cd. Thus, the second goal of this study was to reveal the fine effects of Cd incorporation, which could manifest themselves in the luminescent properties (including luminescence quantum yields and lifetimes). Finally, the development of new ways for assembling coordination polymers has also been challenging. It is known that, in many cases, stable polynuclear cores are formed due to the self-assembly of metal cations and ligand anions, for example, Fe^III^_2_M^II^ (M^II^ = Mn, Fe, Co, Ni and other) [34,35,36,37,38]. It has recently been shown that the combination of Cd^II^ ions with Eu^III^ leads to the formation of compounds containing LnCd/LnCd_2_/Ln_2_Cd_2_ cores [39,40,41,42,43]. It was attractive to extend this methodology to the synthesis of the Ln^III^-Cd coordination polymers.

In this paper, we report on the synthesis, X-ray diffraction analysis, luminescence properties, as well as experimental and quantum chemical studies of magnetic properties of a series of coordination polymers [Ln^III^Cd_2_(piv)_7_(H_2_O)_2_]_n_·nMeCN, where Ln^III^ = Sm (**1**), Eu (**2**), Tb (**3**), Dy (**4**), Ho (**5**), Er (**6**), Yb (**7**).

## 2. Results and Discussion

### 2.1. Synthesis and Structural Study of Complexes ***1**–**7***

The reaction of cadmium(II) trimethylacetate [Cd(H_2_O)_2_(piv)_2_] with lanthanide(III) nitrates in the ratio Cd:Ln = 2:1 in MeCN produced 1D polymeric complexes [Ln^III^Cd_2_(piv)_7_(H_2_O)_2_]_n_·nMeCN (Ln^III^ = Sm (**1**), Eu (**2**), Tb (**3**), Dy (**4**), Ho (**5**), Er (**6**), Yb (**7**)). Upon addition of a stoichiometric amount of 2,4-lutidine to complex **2**, unchanged starting compound **2** was isolated. In contrast, the use of chelating *N*-donor ligands (1,10-phenantroline (phen), 2,2′-bipyridine (bpy)) caused the destruction of the heterometallic metal core and the formation of complexes [Eu_2_(piv)_6_(bpy)_2_], [Eu_2_(piv)_6_(phen)_2_] [44]. It should be noted that the metal core did not undergo rearrangement upon replacement of the pivalate anion by the dianion of 1,4-naphthalenedicarboxylic acid leading to the formation of a metal-organic framework structure [45].

Complexes **1**–**4**, **6** and **7** were characterized by single-crystal X-ray diffraction analysis. All the compounds were isostructural (Appendix A), so only their general structure is presented and the main geometrical parameters are compared (Table 1). The complexes crystallize as solvates with one acetonitrile molecule per formula unit. In 1D polymeric chains, one can distinguish trinuclear linear fragments {LnCd_2_}, which play the role of monomeric fragments (Scheme 1, Figure 1a). In such units, metal centers are linked by carboxylate bridges from piv anions. The transition from Sm^III^ to Yb^III^ is accompanied by a change in the geometry of the LnO_8_ polyhedron. The coordination environment of the Sm^III^ ion in **1** has the geometry of a biaugmented trigonal prism, the environment of Eu^III^ ion in **2** can be described as both a triangular dodecahedron and a biaugmented trigonal prism with a minimal deviation from an ideal polyhedron among 13 types of known 8-vertex polyhedra (Figure 1b). The geometry of the LnO_8_ polyhedron in **3**, **4**, **6**, **7** corresponds to a triangular dodecahedron (Appendix A). Cadmium metallocentres are bound to oxygen atoms of bridging and chelate-bridging carboxylate groups as well as of one water molecule. The Cd1 ion is located in a pentagonal bipyramidal (*D*_5h_) CdO_7_ environment, where the atoms O1, O2, O5, O6, and O14 are located in the equatorial plane. The geometry of the Cd_2_ ion’s environment (CdO_6_) corresponds to both the trigonal prism (*D*_3h_) and octahedron (*O*_h_). Hydrogen atoms of coordinated water molecules participate in the formation of H-bonds with the O atoms of the carboxylate groups (inside the polymer chain) and the N atom of the MeCN solvate molecule. Thus zigzag polymer chains are formed, the minimal distance between Ln ions (>10 Å) in the crystal corresponds to the distance in the polymer chain, while the shorter distance between Ln ions of neighboring chains is more than 11 Å (Figure 1b,c).

It was previously shown that the combination of Ln^III^ with Zn^II^ ions gives molecular complex [EuZn_2_(piv)_6_(MeCN)_2_] under the same conditions that are used herein for assembling Ln^III^–Cd^II^ coordination polymers [46]. The zinc(II)-lanthanide(III) analog [EuZn_2_(piv)_6_(MeCN)_2_] has a trinuclear metalcore with a central Eu^III^ atom, which is similar to the crystallographically-independent unit of complex **2** (taking into account that positions of Zn^II^ were occupied by Cd^II^ in **2**). All six piv anions in the compound have a bridging type of coordination and bind Eu^III^ to Zn^II^ atoms. Only one acetonitrile molecule is coordinated with both zinc(II) atoms in the terminal position and completes the formation of the molecular compound. The difference in ionic radii of Zn^II^ and Cd^II^ (*r*(Zn^2+^) = 0.88 Å, *r*(Cd^2+^) = 1.09 Å) [47] may be responsible for the difference in the structure of Ln-Zn and Ln-Cd complexes. As a consequence, the cadmium ion can have higher coordination numbers (CN = 5–7) compared to the zinc ion (CN = 4–6), and the Cd^II^ ions can coordinate the oxygen atoms of neighboring trinuclear fragments {LnCd_2_(piv)_6_}. Thus, cadmium ions induce the polymerization of Cd_2_Ln fragments with the formation of polymeric Ln-Cd pivalates. A similar situation was previously observed for complexes [Ln_2_M_2_(pfb)_10_(phen)_2_] (Ln = Eu, Gd, Tb, Dy; M = Zn, Cd; pfb^-^ is the anion of pentafluorobenzoic acid): Ln_2_Zn_2_ complexes had a discrete molecular structure, while the Ln_2_Cd_2_ species were molecular or 1D polymers as a result of changes in the functionality of bridging carboxylate groups and π-π interactions between aromatic fragments (pfb anion and phen ligand) of neighboring tetranuclear fragments [48]. Similarly, 1D polymer [CdEu_2_(pfb)_8_(Etypy)(H_2_O)_2_]_n_ (Etypy = 3-ethynylpyridine) was reported, where Cd^II^ ion adopted coordination number 7 and could be bound in the chain [39].

In all these examples, as in the present study, the high ionic radius of Cd^II^ and its ability to form a large number of coordination bonds at least contributed to the polymerization of Cd*_x_*Ln_2_ (*x* = 1 or 2) moieties and, in some cases, were the main driving force behind the formation of coordination polymers. As expected, molecular complexes with Cd*_x_*Ln*_y_* pivalate cores were formed upon the addition of “capping ligands”, which blocked the coordination sites of Cd^II^ ions and prevented the formation of coordination bonds between these ions and pivalate oxygen atoms from the neighboring units [39,41,43,48,49,50]. Among the heterometallic Cd-Ln carboxylates that do not contain specific capping ligands, only the compound [Cd_2_Eu(bzo)_6_(NO_3_)(MeCN)_2_(THF)_2_] (bzo^−^ = 3,5-di-*tert*-butylbenzoate anion) had a molecular structure, and the coordination positions of the terminal Cd^II^ ions were blocked by coordinated THF and MeCN molecules [51]. Notably, this compound differs from all previous examples like the carboxylate ligand. Most likely, the formation of a molecular compound, in this case, is caused by the lower donor ability of O atoms of bzo^-^ compared to pivalate, however, other effects (such as the energy of the crystal lattice) cannot be excluded.

Complexes **1**–**7** are stable when stored in air, their phase purity and isostructurality were determined using PXRD (Appendix A).

IR spectra of the synthesized complexes **1**–**7** are similar. The observed absorption bands correspond to symmetric (s), asymmetric (as), deformation (δ), and skeleton (γ) vibrations of methyl groups in piv-anions, as well as valence (ν) vibrations of C-H bonds and carboxy groups. These bands are located at the wavenumber ranges 2961–2966 cm^−1^, 2921–2929 cm^−1^, 2864–2903 cm^−1^ (three bands ν(CH_3_)), 1530–1563 cm^−1^ (ν_as_(COO)), 1480–1486 cm^−1^ (δ_as_(CH_3_)), 1416–1427 cm^−1^ (ν_s_(COO)), 1377–1380 cm^−1^, 1360–1365 cm^−1^ (two bands δ_s_(CH_3_)), 1220–1228 cm^−1^ (γ(CH_3_)). The difference Δν = ν_as_(COO) − ν_s_(COO) is 135–166 cm^−1^, which indicates the presence of both coordination types of the carboxylic groups—Chelate and bridging [52], which is consistent with the data of X-ray diffraction analysis.

### 2.2. Photoluminescence Properties of ***2**–**4***

Figure 2, Figure 3 and Figure 4 display the excitation and emission spectra of polycrystalline samples of compounds **2**–**4**, measured at ambient temperature. Compounds exhibit bright luminescence in the red (**2**) or green (**3** and **4**) regions of the spectrum. The absence of broad bands of the ligands in the excitation spectra indicates the inability of sensitization through ligands. The high triplet level of the piv ligand (determined earlier as 27,470 cm^−1^ [46] does not promote the energy transfer to lanthanide ions, nevertheless, the characteristic red and green emission of Eu^3+^ and Tb^3+^ ions can be observed with the naked eye under ultraviolet excitation. 

The emission spectrum of complex **2** demonstrates characteristic narrow bands at 579, 590, 616, 651, and 700 nm associated with the ^5^D_0_–^7^F_J_ (J = 0–4) transitions of Eu^3+^, respectively. The most intense ^5^D_0_-^7^F_2_ transition, named hypersensitive, is highly dependent on changes in the Eu^III^ ion environment. In contrast, the probability of the ^5^D_0_–^7^F_1_ magnetic dipole transition in the first approximation can be considered constant, therefore, this transition is often used as a measure of the luminescence intensity. The ratio of the integrated intensities I(^5^D_0_–^7^F_2_)/I(^5^D_0_–^7^F_1_) is 4.8 for **2** and indicates the absence of an inversion center at the Eu^3+^ position [53]. The single symmetric line of the ^5^D_0_–^7^F_0_ transition, as well as the monoexponential decay of the luminescence, indicate the presence of a unique crystal site of Eu^III^. These data agree with the single-crystal X-ray diffraction data.

The emission spectrum of **3** (Figure 3) demonstrates the typical Tb^III^ luminescence bands at 486, 542, 585, and 620 nm associated with transitions from the ^5^D_4_ excited state of Tb^3+^ to the ^7^F_J_ multiplets (J = 6–3), respectively. The ^5^D_4_–^4^F_2_ transition demonstrates low intensity at 640 nm. The most intense band corresponds to the ^5^D_4_–^7^F_5_ transition; its intensity is about 62% of the total integrated intensity.

The emission spectrum of **4** consists of three lines at 480, 575, and 665 nm, corresponding to the ^4^F_9/2_–^6^H_15/2_, ^4^F_9/2_–^6^H_13/2,_ and ^4^F_9/2_–^6^H_11/2_ transitions of Dy^3+^, respectively (Figure 4). The ^4^F_9/2_–^6^H_13/2_ transition is dominant; its intensity is about 64% of the total intensity.

The excitation spectra of complexes **2**, **3**, and **4** (Figure 2, Figure 3 and Figure 4) demonstrate a series of narrow bands corresponding to the 4f–4f transitions of lanthanide ions and the absence of the broad absorption bands of the ligands. The excitation spectrum of **3** features a strong broadband with a maximum at ~31,000 cm^−1^, which is absent in the spectra of **2** and **4** and may belong to the 4f–5d parity allowed transition of Tb^III^.

The luminescence decay curves for **2** and **3** are well fitted by monoexponential functions (Table 2). The lifetimes of the metal-centered luminescence were long due to the absence of low-lying energy levels contributing to the depopulation of the ^5^D_0_ (Eu^III^) and ^5^D_4_ (Tb^III^) excited states, as well as the lack of efficient quenchers in the closest surrounding of lanthanide ions. The ligand environment prevents the coordination of solvent molecules to the lanthanide ion, which leads to a low rate constant of the non-radiative decay of the Eu^III^ excited state.

The intrinsic quantum yield ($Q_{Ln}^{Ln}$) calculated for **2** turned out to be higher than for Eu(NO_3_)_3_·6H_2_O [54], and is comparable with that of aromatic carboxylate complexes of Eu^III^ (Table 2). The lifetimes of the excited state (*τ*_obs_) of complexes **2** and **3** are comparable to those for the Ln^III^-Zn and Ln^III^-Cd heterometallic complexes with aromatic carboxylate ligands, as well as for the Ln^III^ pivalate complexes with coordinated aromatic N-donors, in which radiative decay prevails over nonradiative (for Eu^III^-containing complexes) (Table 2). These results confirm that the presence of “antenna” ligands and the absence of water molecules in the coordination sphere of the Ln^III^ ion contribute to an increase of the lifetime, effective sensitization of emission, as well as a high quantum yield.

### 2.3. Magnetochemical Measurements and Modeling

Direct current (*dc*) magnetic susceptibilities in 5000 Oe dc-field were measured for compounds **3** (TbCd_2_), **4** (DyCd_2_), **5** (HoCd_2_)**,** and **7** (YbCd_2_) in the temperature range of 2–300 K (Figure 5). The χ_M_*T* values at room temperature were 11.6, 14.0, 13.3, and 2.4 cm^3^·K·mol^−1^ for **3**, **4**, **5,** and **7**, respectively. These values are in a good agreement with the expected values for one isolated Tb^III^ (*S* = 3, *L* = 3, *g* = 3/2, ^7^F_6_, χ*T* = 11.82 cm^3^·K·mol^−1^), Dy^III^ (*S* = 5/2, *L* = 5, *g* = 4/3, ^6^H_15/2_, χ*T* = 14.17 cm^3^·K·mol^−1^), Ho^III^ (*S* = 2, *L* = 6, *g* = 5/4, ^5^I_8_, χ*T* = 14.07 cm^3^·K·mol^−1^), and Yb^III^ (*S* = 1/2, *L* = 3, *g* = 8/7, ^2^F_7/2_, χ*T* = 2.57 cm^3^·K·mol^−1^) [64]. With a temperature decrease, the χ_M_*T* values for **4** remain almost constant up to 100 K and then decrease with further lowering the temperature, reaching a value of about 8.5 cm^3^·K·mol^−1^ at 2 K. The χ_M_*T* values for **3**, **5**, and **7** decreases with decreasing temperature over the entire temperature range, reaching minimum values of 6.3, 4.8 and 0.64 cm^3^·K·mol^−1^, respectively, at 2 K. Such magnetic behavior may be the result of depopulation of the excited Stark sublevels, corresponding to a number of crystal-field (CF for **3**, **6**) or Kramers (KD for **4** and **7**) doublets [65]. χ_M_*T* vs. *T* dependencies of **3**, **4**, **5**, and **7** could be simulated using a model based on the Hamiltonian (1), similarly to reported cases [65,66,67,68,69].
(1)$$\hat{H}=\Delta_{\mathrm{Ln}}J_{z}^{2}$$
where Δ_Ln_ is the parameter of electronic levels splitting by crystal field of axial symmetry, and *J*_z_ is the operator of the full angular momentum of Ln^III^ ion. Consideration of the molecular field (*z**J* term) [69] improved the fit in all cases. The simulation was performed using Mjollnir software [69,70,71,72] as described previously [37].

The best correspondence between the experimental and calculated χ_M_*T* vs. *T* curves was achieved at Δ_Tb_ = −15.9 cm^−1^, *zJ* = −0.04 cm^−1^ for **3** (*g*_Tb_ = 3/2, *R*^2^ = 6.1 × 10^−4^, which is defined as *R*^2^ = Σ(χ*T*_calc_ − χ*T*_obs_)^2^/Σ(χ*T*_obs_)^2^), Δ_Dy_ = 3.9 cm^−1^ for **4** (*g*_Dy_ = 4/3, *R*^2^ = 6.7 × 10^−4^), Δ_Ho_ = 10.0 cm^−1^, *zJ* = −0.18 cm^−1^ for **5** (*g*_Ho_ = 5/4, *R*^2^ = 3.1 × 10^−3^) and Δ_Yb_ = 125 cm^−1^, *zJ* = −0.73 cm^−1^ for **7** (*g*_Yb_ = 8/7, *R*^2^ = 2.8 × 10^−4^) (Appendix A). The values of Δ_Ln_ are quite close to the values, reported previously for Dy [37] and Ho [69], however, the correspondence between the experimental and calculated χ_M_*T* vs. *T* curve for **5** was not very good (*R*^2^ > 3 × 10^−3^). Thus, simple model based on the above Hamiltonian cannot describe magnetic properties of these two complexes adequately.

We also used the SINGLE_ANISO code and the results of SA-CASSCF/SO-RASSI calculations for model clusters **4m** and **7m** to simulate the χ_M_*T* temperature dependences for **4** and **7** (see Section 3.6, Appendix A). Indeed, our ab initio modeling, which takes into account the temperature-dependent population of Kramers doublets of isolated model clusters (Table 3), is in good agreement with experiment for **7** and in reasonable agreement—For **4** (Figure 5).

To investigate the magnetization dynamics, alternating current (ac) magnetic susceptibility measurements were performed for polycrystalline samples of **3**–**5** and **7**. The studied complexes did not demonstrate the presence of slow magnetic relaxation in the zero magnetic field. The application of dc-magnetic field made it possible to observe non-zero values of the imaginary component of magnetic susceptibility for **4** and **7**. Such a change in the magnetic behavior in the presence of an external magnetic field usually indicates a rather strong contribution of quantum tunneling of magnetization (QTM) to the relaxation process, which significantly accelerates the rate of relaxation. For **3** and **5**, the deviation from zero of the χ″ value was within the instrument error range even in non-zero dc-magnetic fields (Appendix A). The high efficiency of QTM in complexes **4** and **7** is due to their insufficiently high magnetic axiality, as evidenced by the rather high values of *g*_x_, *g*_y_ (Table 3), which leads to large values of the corresponding matrix elements between components of KDs of the transversal magnetic moment (Appendix A) [64].

For the most effective neutralization of the QTM effect, it is necessary to determine the optimal field at which the relaxation time is the longest. Measurement of the ac-magnetic susceptibility in the dc-field range from 0 to 5000 Oe at 2 K made it possible to determine the optimal field value, at which the maximum values of the imaginary component of the ac-susceptibility are shifted to the lowest frequencies; these field values were equal to 1000 Oe for both **4** and **7** (Figure 6).

The results of measuring the ac-magnetic susceptibility of complex **4** in the optimal dc-field are shown in Figure 7. The relaxation time τ_0_ = 1/2πν_max_ was determined by processing the dependences χ′(ν) and χ″(ν) using the generalized Debye model. Approximation of the high-temperature part of the τ(1/*T*) dependence using the Arrhenius equation (Orbach relaxation mechanism, τ_Orbach_ = τ_0exp_{Δ*E*_eff_/*k*_B_*T*}) led to an evaluation of the effective energy barrier of magnetization reversal and the characteristic relaxation time, Δ*E*_eff_/*k*_B_ = 15 K and τ_0_ = 5.6·10^−7^ s, respectively (Figure 8).

For compound **7**, the maxima on the frequency dependence of χ″ were observed in an optimal field of 1000 Oe in the temperature range 2–4 K (Figure 6). The τ values were evaluated as in the previous case. Δ*E*_eff_/*k*_B_ and pre-exponential factor (τ_0_) were determined by approximation of the high-temperature data using the Arrhenius law and found to be 13 K and 5.7 × 10^−7^ s, respectively (Figure 9). To approximate the experimental data in the entire temperature range the sum of Orbach, Raman, and direct mechanisms were used. The following values of the parameters were obtained: Δ*E*_eff_/*k*_B_ = 7.2 K, τ_0_ = 7.1 × 10^−6^ s, *C*_Raman_ = 2.8 K^−7^s^−1^, *n*_Raman_ = 7, *A*_direct_ = 1.8 × 10^−9^ s^−1^ Oe^−4^ K^−1^, *n*_direct_ = 4, *R*^2^ = 0.99996, where *R*^2^ was determined using formula.
(2)$$R^{2}=\frac{Explained\mathrm{var}iation}{Total\mathrm{var}iation}=\frac{TSS-RSS}{TSS}=1-\frac{RSS}{TSS}$$

On the other hand, it should be borne in mind that the Orbach process does not always contribute to the relaxation process, which has already been observed for some Er and Yb complexes [73]. The absence of such a contribution can be evidenced by the ab initio prediction of the energy of the first excited state, which exceeds the effective barrier by two times for **4** and by an order of magnitude for **7** (Table 3), as well as by sufficiently large values of τ_0_. The relaxation times characteristic of the over-barrier magnetization reversal corresponding to the Orbach mechanism should be ~10^−10^–10^−12^ s. For **4** and **7**, the values of τ_0_ are very far from these limits. Thus, we tried to explain the experimental data excluding the Orbach regime from the analysis of relaxation processes according to ref. [74].

For **4**, the sum of Raman (τ_Raman_^−1^ = *C*_Raman_*T^n^*^_Raman^) and temperature independent (τ_QTM_^−1^ = *A*) terms described well the experimental τ vs. 1/*T* dependence over the entire temperature range (Figure 8, red line) with the following parameters: *C*_Raman_ = 31 ± 2 K^−*n*_Raman^s^−1^, *n*_Raman_ = 5.00 ± 0.06, *A* = 2970 ± 40 s^−1^, *R*^2^ = 0.99996. Furthermore, the data fitting by the sum of the direct and Raman relaxation mechanisms also provided satisfactory results with the following parameters: *A*_direct_ = (1.700 ± 0.005) × 10^−9^ K^−1^Oe^−4^s^−1^, *n*_direct_ = 4 (fixed for Kramers ions), *C*_Raman_ = 11 ± 3 K^n_Raman^s^−1^, *n*_Raman_ = 6.1 ± 0.2, *R*^2^ = 0.99986. The fits with other sets of mechanisms led to over-parameterization.

Similar to the case of **4**, the approximation of the temperature dependency of the relaxation time for **7** (Figure 10) was performed with sufficient confidence, taking into account the direct and Raman mechanisms of relaxation with the following parameters: *C*_Raman_ = 1040 ± 100 K^−n_Raman^s^−1^, *n*_Raman_ = 2.9 ± 0.1, *A*_direct_ = (7.4 ± 1.0) × 10^−12^ K^−1^Oe^−4^s^−1^, *n*_direct_ = 4, *R*^2^ = 0.9959. Thus, we propose that the magnetization relaxations in compounds **4** and **7** take place predominantly by the direct and Raman mechanisms.

A large number of Dy^III^ complexes exhibit SMM properties, but the search for conditions leading to the formation of a certain geometry of the coordination environment of a metal ion and its isolation from other paramagnetic ions is a rather difficult task. The proposed method of incorporating Dy^III^ into a polymeric chain of Cd^II^ ions and pivalate ligands made it possible to isolate metal ions from each other, but the geometry of the coordination environment did not favor to slow magnetic relaxation by the Orbach mechanism. The creation of a certain geometry of the coordination environment is apparently governed by the unpredictable action of several factors. For example, diamagnetic dilution of Dy^III^ ions by Zn^II^ in the heterometallic trinuclear {DyZn_2_} complex with a Schiff-base and carboxylate ligands was associated with the formation of DyO_8_ polyhedron with the geometry of square antiprism (Dy···Dy 9.736 Å) [75]. This {DyZn_2_} complex possessed a field-induced slow magnetic relaxation with Δ*E*_eff_/*k*_B_ ≈ 12.3 K (2 kOe) [75]. On the other hand, in the case of 1D-polymer [Dy_2_(piv)_5_(OH)(H_2_O)]_n_ based on tetranuclear fragment {Dy_4_(piv)_6_(μ_3_-OH)_2_} [76], the field-induced slow relaxation of magnetization was revealed using the ac-magnetic data analysis, but the barrier was much lower (Δ*E*_eff_/*k*_B_ ≈ 4.5 K), this lower value was presumably caused by intramolecular exchange interactions between Dy^III^ ions (Dy···Dy 3.790−4.175 Å). The same paper [76] reported on the binuclear complex [Dy_2_(piv)_6_(phen)_2_] (Dy···Dy 5.391 Å), for which the field-induced slow magnetic relaxation was also observed and the barrier was estimated using the Arrhenius equation as ≈ 28.4 K.

A direct comparison of the magnetic properties of heterometallic complexes of lanthanides with Zn^II^ or Cd^II^ cations is hampered by the small number of such complexes with similar geometric parameters. Since the Zn^II^ and Cd^II^ ions are diamagnetic, the difference in their effect on the magnetism of the Zn-Ln and Cd-Ln compounds is exclusively due to their influence on geometric characteristics. As noted above, the larger ionic radius of the Cd^II^ cation and, accordingly, its higher possible coordination numbers in comparison with Zn^II^ are the main sources of differences in geometric characteristics. For the above zinc complexes, in one case, two relaxation pathways were observed, which is quite typical for heterometallic Dy^III^ complexes. The existence of two relaxation pathways can be associated with several asymmetric units and, possibly, with low-temperature isomers/conformers or Ln-Ln interactions, which perturb the electronic structure of some Ln^III^ ions. In the CdLn complexes, we observed only one relaxation pathway, which indicates that, in this particular case, the Cd-based structure was more symmetric and/or rigid in view of the changes caused by temperature (as shown in [37]). In Cd-Ln complexes, the higher separation of Ln^III^ ions may be responsible for their better magnetic isolation and the absence of disturbing interactions. However, there are currently insufficient data to formulate general conclusions.

## 3. Experimental and Computational Details

### 3.1. Materials and Methods

The compounds were synthesized in the air using commercial MeCN solvent (˃99%). Commercial Sm(NO_3_)_3_·6H_2_O (99%), Eu(NO_3_)_3_·6H_2_O (99%), Tb(NO_3_)_3_·6H_2_O (99%), Dy(NO_3_)_3_·5H_2_O (99%), Ho(NO_3_)_3_·5H_2_O (99%), Er(NO_3_)_3_·5H_2_O (99%), Yb(NO_3_)_3_·6H_2_O (99%), were used without additional purification. Starting compound [Cd(H_2_O)_2_(piv)_2_] was synthesized from Hpiv (99%, Merck, Darmstadt, Germany) and Cd(NO_3_)_2_·4H_2_O (>99%, Acros Organics, Waltham, MA, USA) according to the known procedure [77]. Elemental analysis was carried out on an EA1108 Carlo Erba automatic CHNS-analyzer. IR spectra of the compounds were recorded on a Perkin Elmer Spectrum 65 spectrophotometer equipped with a Quest ATR Accessory (Specac, Orpington, UK) by the attenuated total reflectance (ATR) in the range 400–4000 cm^−1^. Luminescent spectra were measured with a Perkin Elmer LS-55 spectrofluorometer.

### 3.2. Synthesis of the Compounds

0.28 mmol Ln(NO_3_)_3_·*x*H_2_O (*x* = 5 for Ln = Dy, Ho, Er and 6 for Ln = Sm, Eu, Tb, Yb) was added to a solution of 0.20 g [Cd(H_2_O)_2_(piv)_2_] (0.57 mmol) in 20 mL of MeCN. The reaction mixture was stirred for 20 min at 80 °C, cooled to room temperature, and filtered. The solution was kept at room temperature and colorless crystals of complexes precipitated after 72 h. The crystals were filtered off, washed with cold MeCN (*t* = −5 °C), and dried in air at *t* = 20 °C.

#### 3.2.1. [SmCd_2_(piv)_7_(H_2_O)_2_]_n_·nMeCN (**1**)

Yield: 0.21 g (63% counting per Sm(NO_3_)·6H_2_O). Calc. for C_37_H_70_NO_16_Cd_2_Sm (%): C 38.0; H 6.1; N 1.2. Found (%): C 38.3; H 6.2; N 1.6. IR (ν, cm^−1^): 3391 m, 2966 m, 2161 w, 1670 w, 1592 m, 1482 s, 1418 s, 1378 s, 1307 m, 1226 s, 1080 w, 1032 w, 900 m, 806 m, 789 m, 603 m, 541 m, 421 m.

#### 3.2.2. [EuCd_2_(piv)_7_(H_2_O)_2_]_n_·nMeCN (**2**)

Yield: 0.28 g (86% counting per Eu(NO_3_)_3_·6H_2_O). Calc. for C_37_H_70_NO_16_Cd_2_Eu (%): C 38.3; H 6.1; N 1.2. Found (%): C 38.1; H 6.7; N 1.0. IR (ν, cm^−1^): 3352 w, 2961 w, 1594 m, 1534 s, 1511 s, 1480 s, 1460 m, 1416 s, 1362 s, 1224 m, 1031 w, 900 w, 805 w, 789 w, 602 m, 562 w, 540 m, 414 w.

#### 3.2.3. [TbCd_2_(piv)_7_(H_2_O)_2_]_n_·nMeCN (**3**)

Yield: 0.26 g (78% counting per Tb(NO_3_)*_3_*·6H_2_O). Calc. for C_37_H_70_NO_16_Cd_2_Tb (%) C 38.0; H 6.0; N 1.2. Found (%): C 37.7; H 6.1 N 1.2. IR (ν, cm^−1^): 3381 w, 2963 m, 2871 w, 1597 m, 1535 s, 1514 s, 1481 s, 1461 m, 1418 s, 1378 s, 1362 s, 1225 s, 1032 w, 939 w, 901 m, 806 m, 789 m, 604 s, 564 m, 542 m, 458 w, 450 w, 440 w, 421 m, 407 m.

#### 3.2.4. [DyCd_2_(piv)_7_(H_2_O)_2_]_n_·nMeCN (4)

Yield: 0.26 g (78% counting per Dy(NO_3_)_3_·5H_2_O). Calc. for C_37_H_70_NO_16_Cd_2_Dy (%): C 37.9; H 6.0 N 1.2. Found (%): C 37.6; H 6.3; N 1.0. IR (ν, cm^−1^): 3337 m, 3256 m, 2961 w, 1704 w, 1612 s, 1563 s, 1486 s, 1428 s, 1378 s, 1360 s, 1200 s, 1030 w, 899 m, 824 w, 809 w, 791 w, 672 w, 598 m, 560 m, 453 w, 418 w.

#### 3.2.5. [HoCd_2_(piv)_7_(H_2_O)_2_]_n_·nMeCN (**5**)

Yield: 0.17 g (55% counting per Ho(NO_3_)·5H_2_O). Calc. for C_37_H_70_NO_16_Cd_2_Ho (%): C 37.8; H 6.0; N 1.2. Found (%): C 37.5; H 6.4; N 1.4. IR (ν, cm^−1^): 2962 m, 1704 w, 1585 m, 1539 s, 1483 s, 1462 m, 1427 s, 1379 m, 1362 s, 1032 w, 903 m, 811 m, 791 m, 594 m, 563 m, 543 m, 492 w, 450 w, 441 w, 419 m.

#### 3.2.6. [ErCd_2_(piv)_7_(H_2_O)_2_]_n_·nMeCN (**6**)

Yield: 0.18 g (53% counting per Er(NO_3_)·5H_2_O). Calc. for C_37_H_70_NO_16_Cd_2_Er (%): C 37.7; H 6.0; N 1.2. Found (%): C 37.8; H 6.2; N 1.5. IR (ν, cm^−1^): 2964 m, 2162 w, 1673 w, 1531 s, 1483 s, 1457 m, 1381 s, 1365 s, 1231 s, 1032 w, 937 w, 902 m, 790 m, 587 s, 566 s, 492 w, 436 w, 419 m.

#### 3.2.7. [YbCd_2_(piv)_7_(H_2_O)_2_]_n_·nMeCN (**7**)

Yield: 0.21 g (62% counting per Yb(NO_3_)·6H_2_O). Calc. for C_37_H_70_NO_16_Cd_2_Yb (%): C 37.6; H 6.0; N 1.2. Found (%): C 37.4; H 6.3; N 1.4. IR (ν, cm^−1^): 2964 m, 2162 w, 1673 w, 1531 s, 1483 s, 1457 m, 1381 s, 1365 s, 1231 s, 1032 w, 937 w, 902 m, 790 m, 587 s, 566 s, 492 w, 436 w, 419 m.

### 3.3. X-ray Diffraction Studies

Single crystal X-ray studies of crystals **1**–**4**, **6** and **7** were carried out on a Bruker Apex II diffractometer equipped with a CCD detector (MoK_α_, λ = 0.71073 Å, graphite monochromator) [78]. A semiempirical adjustment for absorption was introduced for all complexes [79]. Using Olex2 [80], the structures of the compounds obtained were solved by direct methods and refined in the full-matrix least-squares anisotropic approximation using the SHELX software complexes [81]. The hydrogen atoms in the ligands were calculated geometrically and refined in the “riding” model. The crystallographic parameters and the structure refinement statistics are shown in Appendix A. CCDC numbers 2044967 (for **1**), 2044968 (for **2**), 2044970 (for **3**), 2044972 (for **4**), 2044971 (for **6**), 2044969 (for **7**) contains the supplementary crystallographic data for the reported compounds. These data can be obtained free of charge from The Cambridge Crystallographic Data Centre via http://www.ccdc.cam.ac.uk/data_request/cif (accessed on 14 July 2021).

The polyhedron geometry of metals was calculated using the SHAPE 2.1 software [82].

Powder X-ray diffraction data were collected using a Bruker D8 Advance diffractometer (CuK_α_, λ = 1.54 Å, Ni-filter, LYNXEYE detector, geometry reflection).

### 3.4. Magnetic Measurements

Magnetic susceptibility measurements were performed with a Quantum Design susceptometer PPMS-9. This instrument works between 1.8 and 400 K for DC applied fields ranging from −9 to 9 T. For AC susceptibility measurements, an oscillating AC field of 1 or 5 Oe with a frequency between 10 and 10,000 Hz was employed. Measurements were performed on polycrystalline samples sealed in polyethylene bags and covered with mineral oil to prevent field-induced orientation of the crystallites. The paramagnetic components of the magnetic susceptibility χ were determined taking into account the diamagnetic contribution evaluated from Pascal’s constants as well as the contributions of the sample holder and mineral oil.

The magnetization relaxation times τ = 1/2πν_max_ and the α factors, which account for the distribution in relaxation processes, were obtained by fitting the χ′(ν) and χ″(ν) plots using the generalized Debye model (see SI).

### 3.5. Photo-Physical Measurements

Luminescent measurements were performed with a Horiba-Jobin-Yvon Fluorolog FL 3-22 (Horiba Scientific, Kyoto, Japan) spectrometer, which has a 450 W xenon arc lamp as an excitation source for steady state measurements and a 150 W xenon pulse lamp for kinetic experiments. An R-928 PMT tube (Hamamatsu Photonics K.K, Hamamatsu, Japan) was used as a detector. The spectra were corrected for instrumental responses. Lifetimes were measured with the same instrument using a xenon flash lamp. The quantum yield measurements were carried out on solid samples with a Spectralone-covered G8 integration sphere (GMP SA, Renens, Switzerland) under ligand excitation, according to the absolute method. Each sample was measured several times under slightly different experimental conditions. The estimated error for quantum yields was ±10%. All complexes studied were powdered before measurements.

### 3.6. Details of Quantum Chemical Calculations

Coordination polymers were divided into smaller structural fragments of individual spin centers, which then could be treated by ab initio computational methods. Calculations were performed for DyCd_2_ and YbCd_2_ clusters using the truncated XRD geometry with the ^tr^Bu groups substituted by Me groups in the piv ligand (hereinafter referred to as **4m**, **7m**, Appendix A). The SA-CASSCF/SO-RASSI approach [83,84,85], implemented in the MOLCAS 8.2 suite of programs [86], was used for calculations. The ANO-RCC-VTZP relativistic basis sets for lanthanides and oxygen atoms with the smaller ANO-RCC-VDZ for other atoms were employed [87]. The scalar relativistic effects were taken into account using the DKH2 Hamiltonian [88]. For the dysprosium complex, 21 sextet, 128 quintet, and 130 doublet states (the energy region up to ~50,000 cm^−1^) were taken into account with the active space consisted of 9 electrons distributed on 7 f-orbitals. For the ytterbium complex, seven singlet and seven triplet states were accounted with the active space consisted of 13 electrons on 7 f-orbitals.

The *g*-tensors for Kramers doublets, their orientations in the molecular axes, and temperature dependence of the molar magnetic susceptibility were evaluated using results of the SA-CASSCF/SO-RASSI calculations and SINGL_ANISO code [89].

## 4. Conclusions

A series of new LnCd_2_ heterometallic 1D polymers [LnCd_2_(piv)_7_(H_2_O)_2_]_n_·nMeCN (Ln = Sm, Eu, Tb, Dy, Ho, Er, Yb) with pivalic acid anions were synthesized and characterized by various methods. Due to the larger ionic radius of Cd^II^ and its ability to have higher coordination numbers in comparison with Zn^II^, the trinuclear units LnCd_2_ undergo polymerization, forming 1D chains, in contrast to the discrete Zn_2_Ln analogs. The polymers [LnCd_2_(piv)_7_(H_2_O)_2_]_n_·nMeCN are isostructural; according to the single-crystal X-ray data, the geometry of the LnO_8_ polyhedron changes from a biaugmented trigonal prism for Ln = Sm and Eu to a triangular dodecahedron for Ln = Tb, Dy, Er, and Yb. In the polymeric chain, lanthanide ions are isolated from each other by two Cd ions (the minimal Ln···Ln distance is more than 10 Å) and coordinates only the oxygen atoms of the bridging carboxylate groups. This type of the closest coordination environment, where efficient quenching groups are absent, gives rise to higher values of the intrinsic quantum yield of luminescence for **2** (EuCd_2_ core) than for Eu(NO_3_)_3_·6H_2_O, and the quantum yield comparable to that for Eu^III^ aromatic carboxylate complexes. The excited state lifetimes for **2** and **3** (EuCd_2_ and TbCd_2_ cores) are comparable to the similar lifetimes of the Ln-Zn and Ln-Cd heterometallic carboxylates. Complexes **4** and **7** (DyCd_2_ and YbCd_2_ cores) exhibit the properties of field-induced SMMs. Based on the analysis of the relaxation of magnetization and the results of high-level ab initio calculations, the contribution of the Orbach relaxation mechanism was excluded. The sum of the Raman and QTM mechanisms was dominant in the magnetization relaxation for **4**, and the sum of the direct and Raman processes was the dominant relaxation mechanism in the case of **7**.

## Data Availability

The data presented in this study are available in this article and Appendix A.

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
