# Peer review of "Cadmium-Inspired Self-Polymerization of {LnIIICd2} Units: Structure, Magnetic and Photoluminescent Properties of Novel Trimethylacetate 1D-Polymers (Ln = Sm, Eu, Tb, Dy, Ho, Er, Yb)"

_molecules, 2021, doi:10.3390/molecules26144296_

Round 1

Reviewer 1 Report

The manuscript under consideration by Shmelev et al report Cadmium inspired self-polymerization of Ln111Cd2 . A series of heterometallic carboxylate 1D polymers LnCd2(piv)7(H2O)2]n·nMeCN (Ln = Sm, Eu, Tb, Dy, Ho, Er Yb) with pivalic acid anion were synthesized and characterized for their structure, magnetic and photolumiscent properties.  The larger radius of Cd2 compared to Zn2, isostructural 1D polymers were obtained.  It is shown that lanthanide ions are isolated by two Cd ions and a coordinate oxygen atom bridging the carboxylate group. Hence, high intrinsic quantum yield of luminescence is obtained that is comparable to aromatic carboxylate complexes of EuIII. The topic under study is important and research in this detection is relevant. In principle, manuscript is well-written and organized. I would recommend the publication the current manuscript after minor revisions listed below.

  1. Narrative is not built appropriately and should be supported by latest references, clearly describing the information obtained in this study that are not already present in the literature. I believe references are there, authors should rephrase and amend the text for better build-up of their point.
  2. Language should be polished well before publishing.
  3. Number of references are too many for a research paper. Authors should consider reduction in number of refences and give only appropriate and latest references.

Author Response

Question 1) Narrative is not built appropriately and should be supported by latest references, clearly describing the information obtained in this study that are not already present in the literature. I believe references are there, authors should rephrase and amend the text for better build-up of their point.

Comment: We took into account this comment and did some changes.

Question 2) Language should be polished well before publishing.

Comment: The manuscript was corrected in accordance with the recommendation. 

Question 3) Number of references are too many for a research paper. Authors should consider reduction in number of refences and give only appropriate and latest references.

Comment: Thank you for the comment. We have reduced self-citation, but we believe that the work of other authors is still worth citing. The final version contains 90 references.

Reviewer 2 Report

This work is quite interesting and will certainly attract many readers. However, I think a minor revision is needed for several reasons.

  1. Page 1, please change “electrochomism” to “electrochromism”.

Page 2, please change “envioronment” to “environment”.

Author Response

Question 1)  Page 1, please change “electrochomism” to “electrochromism”.

Comment: The manuscript was corrected in accordance with the recommendation.

Question 2) Page 2, please change “envioronment” to “environment”.

Comment: The manuscript was corrected in accordance with the recommendation.

Reviewer 3 Report

In this paper, Kiskin reports the synthesis and characterization of several cadmium-containing heterometallic carboxylate 1D polymers with various transition metals. A wide range of molecular and electronic characterization has been carried out such as magnetic and photoluminescent properties. The computational study further demonstrated their hypothesis of magnetic properties. I think these results are interesting and I recommend acceptance, subject to the following, minor corrections:

1 In the synthesis of complex 1-7, Figure 1 only shows the crystal packing. Can they provide more information about the complex? Such as ChemDraw structures of these complexes.

2 In the discussion of complexes (Page 4), the authors mention that “combination of Ln(III) with Zn(II) ions gives molecular complexes”. Can they give some comparison of these complexes in the paper to the Zn(II) complexes?

3 In the magnetic properties section, can they provide a brief description of how these Cd complexes are different from reported Zn complexes?

Author Response

Question 1) In the synthesis of complex 1-7, Figure 1 only shows the crystal packing. Can they provide more information about the complex? Such as ChemDraw structures of these complexes.

Comment: The manuscript was corrected in accordance with the recommendation. Scheme 1 was added.

Question 2) In the discussion of complexes (Page 4), the authors mention that “combination of Ln(III) with Zn(II) ions gives molecular complexes”. Can they give some comparison of these complexes in the paper to the Zn(II) complexes?

Comment: The manuscript was corrected in accordance with the recommendation. Short description of the {EuZn2} pivalate complex was added.

Question 3) In the magnetic properties section, can they provide a brief description of how these Cd complexes are different from reported Zn complexes?

Comment: As we noted in the manuscript there is only a small number of complexes containing d10-elements that were investigated by magnetochemistry methods in details. Unfortunately, magnetic properties of LnZn2-carboxylate complexes were not described early. Therefore, we compared our results with the magnetic properties of known Schiff-base DyZn2-complex (the last paragraph of the section 2.2.).